# Design of a treatment pathway for insomnia in prison settings in England: a modified Delphi study

Lindsay H Dewa,[1,2] Lamiece Hassan,[3] Jenny Shaw,[4] Jane Senior[4]

[1]NIHR Imperial Patient Safety Translational Research Centre, Imperial College London, London, UK
[2]School of Public Health, Imperial College London, London, UK
[3]Division of Informatics, Imaging and Data Sciences, The University of Manchester, Manchester, UK
[4]Offender Health Research Network, The University of Manchester, Manchester, UK

**Correspondence to**
Dr Lindsay H Dewa;
l.dewa@imperial.ac.uk

## ABSTRACT

**Objective** Insomnia is highly prevalent in prisoners and is a risk factor for poor mental well-being, depression, suicidality and aggression, all common concerns in this vulnerable population. Improving sleep management options in prison offers the potential to impact positively on a number of these common risk factors. The study aim was to design a treatment pathway for insomnia in prisons informed by stakeholders with professional or lived experience of insomnia and prison-based interventions.

**Design** A modified Delphi technique, adapted to the stakeholder (either receiving controlled feedback online or face to face on a series of statements), was used over three rounds to gain consensus on a final treatment pathway design.

**Participants** Academic sleep researchers, prison staff and prisoners were invited to develop the treatment pathway.

**Results** Fifteen stakeholders took part in round 1 and thirteen in round 2. There were six statements of contention that comprised concerns over the inclusion of sleep observations, sleep restriction therapy and promethazine. Consensus was high (>80%). Thirteen stakeholders agreed the final pathway in round 3. The final treatment pathway comprised a standardised stepped-care approach for insomnia in prison populations. The pathway resulted in five main stages: (1) transition from community; (2) detection and assessment; (3) treatment for short-term insomnia; (4) treatment for long-term insomnia and (5) transition from prison to community or another establishment.

**Conclusions** The treatment pathway is designed to promote early detection of insomnia, potentially reducing unnecessary prescriptions and medication trading, misuse and diversion in the prison setting. It should make a substantial difference in reducing the large number of sleep complaints and positively impact on prisoners, staff and the prison environment. Specifically, improving sleep should have a positive impact on prisoners' mental and physical well-being and aid smooth running of the prison.

## INTRODUCTION

Good sleep contributes positively to overall health, well-being, social functioning and quality of life.[1] Around a third of the general population experience insomnia at some point in their lives[2] ; however, recent figures show prisoners are at least twice as likely to

### Strengths and limitations of this study

► The first study to codesign a treatment pathway for insomnia in prison, using feedback from a diverse group of academic sleep researchers, prison staff and prisoners. Men and women residing in the prison were included and inputted throughout the research process.

► The pathway informs the development of an integrated, triangulated and coordinated approach to managing insomnia in a prison environment that centres on self-management, peer support and psychological therapy as opposed to early or routine use of hypnotic medication.

► A limited number of stakeholders were included in the pathway development; therefore, results may not have been wholly representative of a full range of stakeholders, particularly across geographical location and different prison types.

► Despite the acceptability of the pathway by stakeholders in this study, a cultural shift, endorsement and buy in from prisoners, staff and practitioners will be needed, to ensure the success implementation of the pathway in practice.

have insomnia and the majority have poor sleep quality.[3–5] This is problematic because insomnia has been shown to significantly contribute to poor cognitive functioning, depression, suicide, emotional dysregulation, aggression and lack of treatment engagement, all of which may affect the safe running of the prison, an individual's rehabilitation and, ultimately, the likelihood of reoffending.[6–9]

National Institute for Health and Care Excellence (NICE) guidelines for treating short-term insomnia recommend non-pharmacological interventions in the first instance followed, if necessary, by a short course of hypnotic medication.[10] For long-term insomnia (symptoms for >4 weeks), medication is not recommended, instead cognitive behavioural therapy for insomnia (CBTi) is considered the gold standard treatment,[10 11] in line with international guidance.[11 12] However, in reality, prisons use a

variety of different approaches and do not seem to follow these guidelines.[13 14] For example, in England and Wales, a recent survey reported that prisons do not routinely implement NICE guidance for managing insomnia.[14] While sleep hygiene education (ie, advice in behaviours that encourage good sleep) was commonplace, offered in 94% of establishments, only one prison from a sample of 84 offered CBTi.[14] Furthermore, hypnotics were commonly prescribed to treat long-term insomnia, notwithstanding guidance advising against their use.[14]

This study was followed up with in-depth qualitative interviews to further understand staff and prisoners' experiences of insomnia treatment and to explore the future direction of insomnia management in prison (L Dewa, unpublished data, 2016). Staff indicated that prisoners should take personal responsibility for managing their insomnia while prisoners suggested structured peer support would be a valuable approach. This current lack of a standardised, stepped-care and evidence-based approach to treatment has both individual well-being and financial implications, including costs to the National Health Service (NHS) due to increased prescribing and greater frequency of healthcare utilisation. Additionally, the lack of adherence to NICE treatment standards is at odds with the now long-standing policy that healthcare provision for prisoners should be equivalent to that provided to the wider community.[15 16]

There is, therefore, a need for insomnia management in prisons to be at least equal to the wider community, so as not to further increase inequalities. Additionally, rationalising the prescription of hypnotics and the introduction of evidence-based psychosocial treatments, with the objectives of improving well-being and lessening health inequalities is a requisite. This study employed a modified Delphi study to design a treatment pathway for insomnia in prisons informed by stakeholders with professional or lived experience of insomnia and prison-based interventions.

## METHODS

A modified Delphi technique was used to develop a treatment pathway for insomnia in prison (figure 1). The

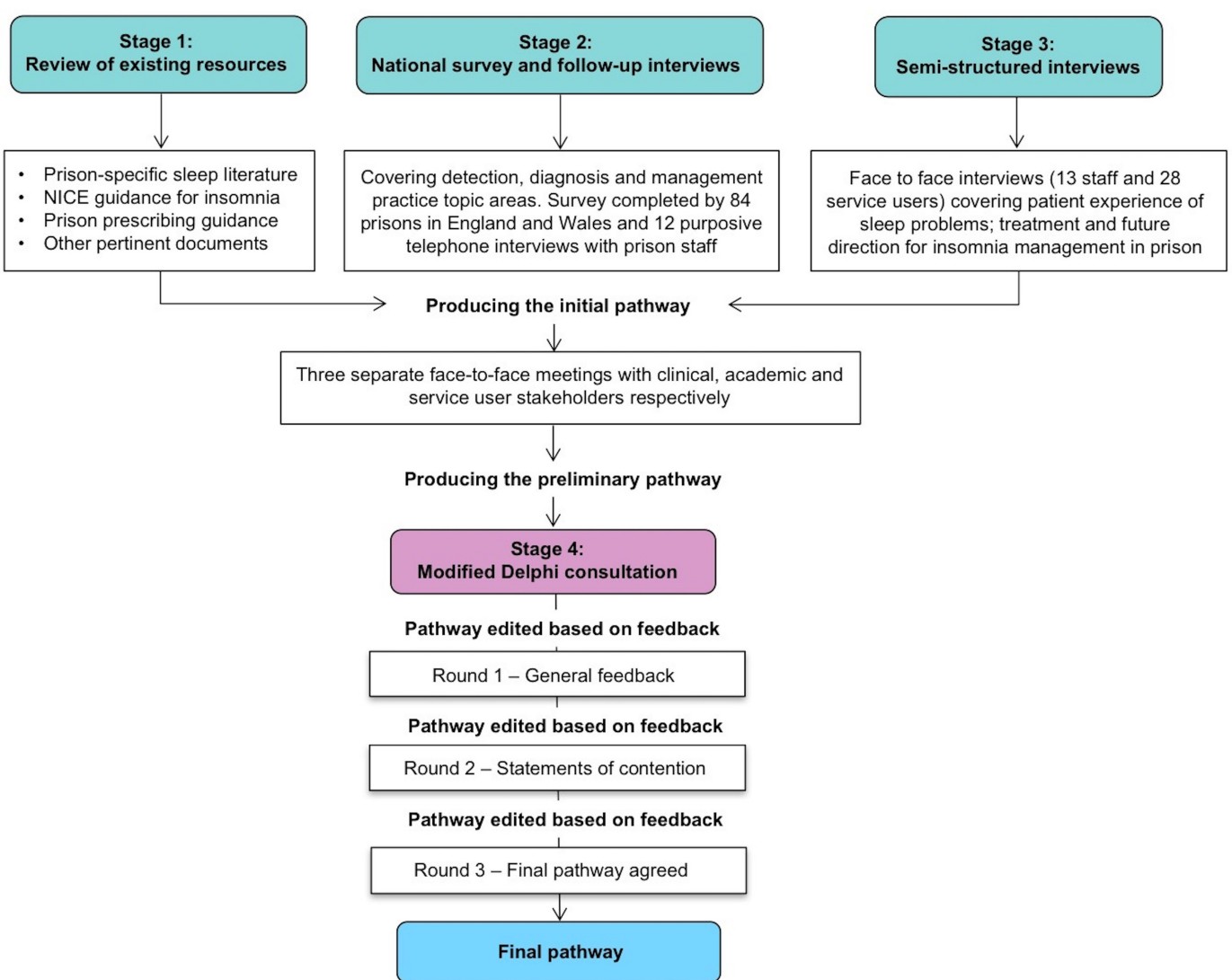

**Figure 1** Stages performed to produce the final pathway. NICE, National Institute for Health and Care Excellence.

Delphi method is an iterative process in which a group of expert stakeholders come to a structured consensus view on the content, style and focus of the subject under discussion, in this instance the insomnia treatment pathway, through a number of rounds of controlled feedback.[17] There is no mandated number of rounds required to complete a Delphi consultation; the process is repeated until consensus is achieved. However, having at least two rounds of feedback is commonplace, with some studies using three.[18–20] Delphi studies have been successfully conducted to improve healthcare guidance in a number of settings, including similar pathway developments in the health and criminal justice systems.[21–25]

To produce a preliminary pathway, current treatment guidelines for insomnia that used a stepped-care approach were reviewed,[10] and findings from the authors' earlier research were triangulated. This included a systematic integrative review of published literature on insomnia in prison[3]; learning from a questionnaire survey and in-depth qualitative interviews with prison health and discipline staff and prisoners about the current treatment of insomnia (L Dewa, unpublished data, 2016)[14]; and a prevalence study of rates of short-term and long-term insomnia in a sample of English prisons.[5] The preliminary pathway then went through an initial review by a convenience sample of academic sleep researchers, prison staff and prisoners. Stakeholders were invited to contribute based on their professional or lived experience of insomnia and prison-based interventions, and availability: academic sleep researchers, because the lead author (LHD) was aware of their expertise in both managing insomnia as a healthcare professional and their research that examined insomnia treatment, and specific prison staff because they had experience in delivering, implementing and evaluating prisoner health issues, including sleep problems. A monthly prisoner health representatives' consultation meeting was identified as a way to purposively identify potential stakeholders. This group had lived experience of insomnia in prison, had previously asked for help for sleep problems in prison and were aware of the range of treatments currently available in prison. Subsequently, the lead author felt, overall, this collective expertise was adequate to proceed with the initial review of the pathway. Traditionally, in a Delphi study, experts do not know each other, do not work or live together and usually respond to consensus rounds via email or other electronic means. This was not possible for prisoners; therefore, the Delphi technique was modified; due to a lack of access to electronic communication in prison and, to fit within the resources of the study, the lead author (LD) met with a group of women prisoners together in prison over a series of three face-to-face meetings.

The degree of consensus was calculated using the percentage of individuals who agreed (either totally agree or somewhat agree) with statements. Feedback from stakeholders was mainly qualitative and therefore analysed in NVivo V.10.[26] Responses were characterised according to emerging themes, with similar responses grouped together to determine levels of consensus. Amendments to the pathway were made after each round of feedback and stakeholders were provided with the revised pathway along with a summary of the results from the previous Delphi round and several new questions/topics to be considered. The provision of summary results allowed stakeholders the opportunity to re-evaluate their views in the wider context and further explore areas of significant disagreement.

### Delphi round 1
Stakeholders were asked to give their initial overall view of the pathway and any omissions/redundant areas via email. Prisoners gave feedback in a face-to-face meeting. This was unstructured to allow independent and novel thoughts to be gathered without limitation to any particular parts of the pathway.

### Delphi round 2
Round 2 adopted a more directive approach. Stakeholders were provided with summarised feedback from round 1 and the resultant changes to the pathway. They were asked to score a series of statements on a 4-point Likert scale (totally agree, somewhat agree, somewhat disagree, totally disagree) to indicate the extent to which they agreed or disagreed with each one. They were also asked to make additional comments on any remaining contentious issues. Consensus was achieved if there was at least 70% agreement (totally agree or somewhat agree) across each statement, in line with guidance.[27]

### Delphi round 3
Consensus was achieved for all statements; therefore, stakeholders were only asked to score the one statement: 'You are happy with the final pathway'. Delphi rounds then ceased.

### Patient and public involvement
The aim of this study was informed by previous research, which has found that insomnia is both highly prevalent and a source of dissatisfaction among people in prison.[3 5 14 28] On the advice of prisoners and prison staff, rather than having a separate involvement group for people in prison, we chose to modify the Delphi design to incorporate prisoner perspectives as a complementary source of insight and expertise in designing the pathway, alongside those of professionals. The resulting preliminary pathway has since been disseminated via stakeholder groups, organisations and publications relevant to both prisoners and ex-offenders and plans have been put in place to enable involvement as part of an ongoing trial of the pathway.

## RESULTS
### Producing the preliminary pathway
Two academic sleep researchers, three prison healthcare staff (two from a male prison holding unconvicted men

and those serving shorter sentences and one male prison for these part way through longer sentences) and six prisoners (from a female prison) reviewed the initial pathway. It was structured to include a short screen for insomnia, using the Sleep Condition Indicator (SCI),[29] an initial assessment (potential causes of insomnia, sleep care plan, etc), and short-term (medication) and long-term treatment (individual components of CBTi) options. Stakeholders described the pathway as comprehensive and detailed, but vast. There were concerns over the medicalisation of insomnia and that general practitioners (GP) were given too much responsibility to detect, diagnose and treat the insomnia. There was an acceptance that insomnia was a widespread issue so help should be available and ongoing, and that people might need different interventions, depending on the duration of insomnia symptoms and/or stage/length of custody.

The research team then made several initial modifications to the pathway, reducing it in size and layout orientation, to increase linearity and to specifically focus on insomnia rather than including additional conditions that were comorbid with insomnia, such as depression. Early interventions were redirected away from GP services towards Improving Access to Psychological Services (IAPT) and primary care services more widely. Continuing support was considered in the form of regular reviews of treatment progress, a prescribed sleep pack (a combination of non-medical items to help promote sleep, eg, ear plugs, Horlicks/hot chocolate and eye mask) and access to trained peer workers, available to give continuous support on residential wings. The revised preliminary pathway was then considered in the Delphi rounds.

### Producing the final pathway
Two academic sleep researchers, two healthcare managers and one GP and three prisoners from the initial group continued into the Delphi rounds and were joined by additional relevant stakeholders (table 1).

### Delphi round 1
Prisoners indicated the pathway was easy to understand, that points raised in the initial discussions had been addressed and that there was welcome scope within it for self-management. Several stakeholders, including the prisoners, expressed concern about having hypnotic medication as part of the pathway at all. Reasons for this included: security concerns; patients' likely gravitation towards medication instead of non-pharmacological intervention due to histories of substance misuse and, in the case of one medication in common current usage, promethazine, no evidence of its effectiveness. Academic sleep researchers highlighted limited evidence for mindfulness and sleep hygiene education and that there was currently too much emphasis on the supposed efficacy of sleep hygiene. In contrast, while the individual components of CBTi were included separately, the entire CBTi package was not included and the academic sleep researchers agreed that this was a limitation. Others agreed that self-directed help

**Table 1** Retention of stakeholders by meeting and Delphi round

| Stakeholder | Initial group meeting | Delphi Round 1 | Round 2 | Round 3 |
|---|---|---|---|---|
| Academic sleep researcher | 2 | 3 | 3 | 3 |
| Prisoner | 3 | 3 | 2 | 2 |
| GP | _ | 2 | 1 | 1 |
| Healthcare manager | 2 | 2 | 2 | 2 |
| Primary care manager | _ | 1 | 1 | 1 |
| Psychologist | _ | 1 | 1 | 1 |
| IAPT staff member | _ | 1 | 1 | 1 |
| Mental health lead | _ | 1 | 1 | 1 |
| Prison governor | _ | 1 | 1 | 1 |
| Total | 7 | 15 | 13 | 13 |

GP, general practitioners; IAPT, Improving Access to Psychological Services.

(eg, expectation management, sleep diaries, sleep packs, exercise in cell, etc) was needed. While all stakeholders agreed insomnia screening, using the SCI was important they disagreed on who should do it, with some favouring a trained peer worker as a stand-alone event and others preferring it to be undertaken by a healthcare professional as part of existing well-man/woman assessments, completed soon after reception into custody.[i] Finally, transition periods (into and out of custody) were also highlighted as important times when sleep support was especially needed.

As a result of this feedback, the role of hypnotic medication was changed to only be offered when an acute stressor was present. Although NICE does not recommend promethazine because of lack of evidence of its effectiveness, the drug is endorsed in the *Safer Prescribing in Prison* guidance designed for the prison environment.[10 30] The entire CBTi treatment package replaced the individual components. It was therefore maintained in the pathway in round 1. Emphasis on sleep hygiene was reduced and only included within the trained peer worker advice, sleep packs and as a component of CBTi. Self-directed sleep packs comprising information on expectation management,[ii] tailored sleep hygiene and details of the insomnia pathway, were added as a resource primarily available at initial reception into custody but that could be revisited throughout the prison stay. Additional changes included care plan development soon after initial reception; the removal of prison officer sleep observations; and the requirement for the insomnia screening to be completed by trained peer workers.

---

[i] Existing care document that every prisoner receives on entry to prison.
[ii] Normalising the acute sleep disruption as part of the transition to a new sleeping environment, including awareness of prison-related noise.

**Table 2** Consensus on round 2 of Delphi, n=13

| Statement of contention | Consensus |
| --- | --- |
| Sleep observations should not be taken as verification of the sleep problem | 13 (100) |
| Sleep restriction therapy should be included | 13 (100) |
| Promethazine should be excluded | 12* (100) |
| Pharmacological intervention should be used in special circumstances only | 13 (100) |
| Screener for sleep disorders other than insomnia should be included | 13 (100) |
| Trained peer worker will conduct insomnia screen | 11 (85) |

*Total number responded.

### Delphi round 2

Stakeholders agreed that sleep packs were useful; peer support should be included; and reliance on sleep hygiene, as a single component, should be reduced. There was disagreement regarding the inclusion of sleep restriction therapy, a CBTi component (ie, limiting time in bed to the average self-reported sleep time, slowly increasing time in bed as sleep improves); that prisoners should conduct insomnia screening; and the inclusion of promethazine. The majority from round 1 responded in round 2 (n=13; 87%); one prisoner did not continue to round 2 because they no longer wanted to be involved, and the GP did not continue because of unprecedented work commitments. Statements of contention are described in table 2. Consensus was high (>80%) for all statements. Each resolved issue was incorporated into the pathway.

### Delphi round 3

As no further changes were needed and no further contentious issues had arisen, stakeholders were only asked to score the single statement: 'You are happy with the final pathway' on the scale described above. All stakeholders who took part in round 3 agreed on the final pathway (n=13; 100%) and no further rounds were necessary. The final pathway is displayed in figure 2.

### DISCUSSION

### A statement of principal findings

A stepped-care approach that encompasses the most effective but least intensive interventions can help ensure acceptable insomnia care.[31] We have now produced a comprehensive treatment pathway based on research and clinical evidence, further modified by consensus across a range of sleep academics, healthcare professional and prisoner stakeholders. The key steps in the pathway are briefly: (1) transition from community, induction and first period in prison; (2) detection and assessment; (3) treatment for short-term insomnia; (4) treatment for long-term insomnia and (5) transition management to the community or transfer to another establishment.

The overarching principle is that insomnia is detected early in the prison term, followed by timely, intervention(s). The pathway includes a greater emphasis on self-management, peer group involvement and psychological approaches, as opposed to early or routine use of hypnotic medication. The SCI[29] based on DSM-V criteria was included to screen for insomnia because it is simple, short and easy to use, indicating an ease of integration into future prison practice. Hypnotic medication may be offered when acute short-term insomnia is present, accompanied with an acute stressor, for example, when a person has been bereaved or received an unexpected or long sentence. If long-term insomnia is detected in prisoners, CBTi is included as part of their long-term insomnia management in line with guidance.[10 11]

### Strengths and weaknesses of the study

Strengths and limitations are evident. This is the first study to codesign a treatment pathway for insomnia in prison. Furthermore, a key strength is the inclusion of prisoners in all stages of the Delphi consultation process. However, there are also some methodological limitations. First, experts from various fields are often likely to know each other; therefore, responses may have been influenced by others' opinions. Second, levels of expertise were not robustly determined before stakeholders were approached. Verifying eligibility for inclusion might have strengthened the validity of the pathway. Lastly, despite no specific agreement on sample size,[32] a limited number of stakeholders were included; therefore, findings may not be wholly representative of the full range of stakeholders, particularly across geographical location and different prison types, including lower and higher secure prisons and older prisoners (age 65+).

### Implications for clinicians and policy-makers

To be successfully launched in prison, the following are needed: strategic involvement of management, endorsement and buy-in from frontline staff and practitioners; the development of sleep peer support roles and self-management sleep packs; and training of relevant staff and peer workers in sleep management. The pathway should be piloted to ensure its effectiveness; this requires time, stakeholder investment and staff resources. As well as implementing the pathway itself, prisons will also need to consider how to positively influence wider determinates of good sleep, including minimising environmental noise, regulating temperature at night-time and encouraging more purposeful daytime activity.

A cultural shift is needed to implement the treatment pathway in prison through an acceptance by staff and prisoners that hypnotic medication prescriptions would reduce significantly, replaced by self-management and peer involvement. Indeed, this may present barriers to its delivery and generate conflicts at higher and lower levels of management between different types of staff and between staff and prisoners, because they are accustomed to locally derived insomnia protocols and

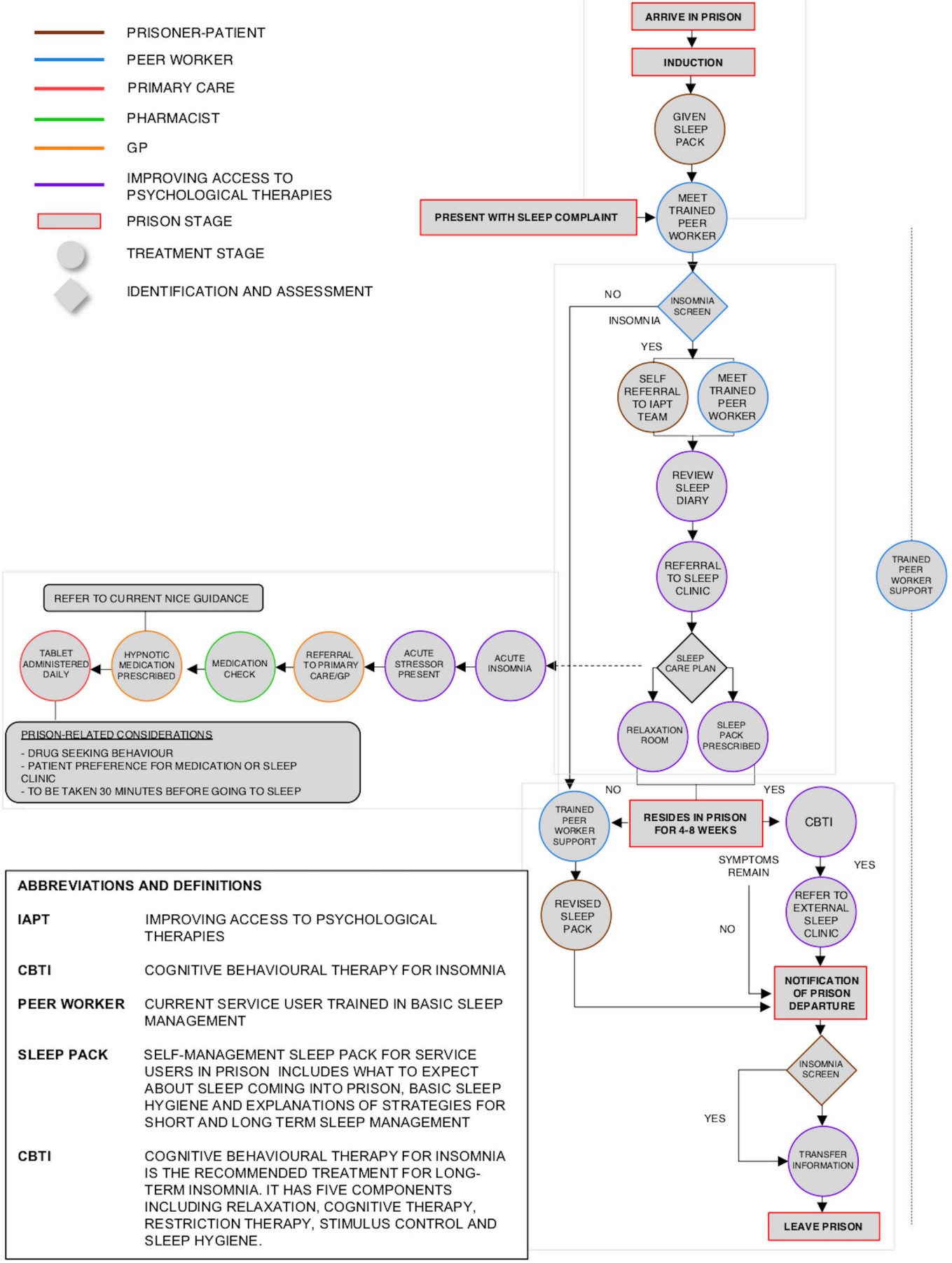

**Figure 2** Final pathway. GP, general practitioners; NICE, National Institute for Health and Care Excellence.

management. Nevertheless, if implemented, we hypothesise the pathway would positively impact on healthcare utilisation rates and help reduce sedative-related prescriptions. Despite these aforementioned barriers, implementation in practice and the emphasis on peer support and self-management should help prisoners take better control of their health and welfare more generally, with wider positive impacts on the prison regime.

## Unanswered questions and future research

In practice, the treatment pathway is likely to require adaptation to different types of prison and will be dependent on the availability of resources, such as trained and supported staff and peer workers. However, it is expected by the authors that the principal direction and content of the pathway will remain the same. In the first instance, a feasibility study should be conducted on the full treatment pathway in a sample of prisons in England to ascertain staff buy-in, acceptability to patients, the costs and practicalities involved in implementing into practice. The feasibility study should use a mixed methods approach comparing the pathway with usual care. It should comprise five main stages:

► Adapting the pathway for the particular prison, in collaboration with prison officers, healthcare staff and prisoners.
► Incorporating the pathway into existing practice and staff training and the development of training and supervision mechanisms for peer workers.
► Testing the feasibility of implementing the pathway in prison.
► Establishing the acceptability of the pathway to staff and prisoners.
► Testing trial processes, including randomisation, recruitment and retention rates and appropriate outcome measurements.

The feasibility of measuring individual-level and prison-level outcomes, cost-effectiveness and ultimate impact on reoffending and wider health and social impacts should also be examined. Specifically, this might include outcome measures such as the impact on insomnia symptoms, general health and well-being, suicidality, mood, productiveness, staff–prisoner and prisoner–prisoner relationships, behaviour and quality of life. In addition, the number of consultations for sleep problems and sedative-related prescriptions would be outcomes of interest.

## Conclusion

We have produced a comprehensive treatment pathway for insomnia with an emphasis on self-management, peer support and psychological approaches, further modified by consensus across sleep academics, healthcare professional and prisoner stakeholders. The codesigned treatment pathway should help to identify, assess and manage insomnia in a population with many sleep complaints. Testing the feasibility of introducing such a pathway would be an essential next step in addressing the significant burden of insomnia on the prison healthcare system and wider prison community and to specifically help guide practitioners to make better decisions for prisoners' sleep and health needs.

**Contributors** LHD drafted and finalised the protocol, designed the study, led the data collection and analysis and interpreted the data. He also drafted the manuscript, approved the final version and is also the corresponding author. LH, JSw and JSr advised on clinical definitions, content interpretation and intellectual content, were involved in drafting and critical appraisal of the manuscript and approved the final version to be published.

**Funding** This work was supported by a Faculty Doctoral Fellowship via a Medical Research Council (grant number: 1233315).

**Competing interests** None declared.

**Patient consent** Not required.

**Ethics approval** The design of the treatment pathway was the last stage of a five study large doctoral grant that was included in the IRAS ethics application; approvals from NHS (REC for Wales; reference: 13/WA/0249) and HM Prison Service ethics committees (NOMS; reference: 2013-208) were obtained. However, the design of the pathway stage was specifically identified as a service improvement project and thus did not require individual ethical approval.

**Provenance and peer review** Not commissioned; externally peer reviewed.

**Data sharing statement** No additional data are available.

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
