## [Reviewer comments · BMJ Open]

ARTICLE DETAILS

TITLE (PROVISIONAL)	The design of a treatment pathway for insomnia in prison settings in England: a modified Delphi study
AUTHORS	Dewa, Lindsay; Hassan, Lamiece; Shaw, Jenny; Senior, Jane

VERSION 1 – REVIEW

REVIEWER	Thomas Fovet MD, PhD, Pôle de psychiatrie, CHU de Lille, 59000 Lille, France and CNRS UMR 9193, laboratoire de sciences cognitives et sciences affectives (SCALab-PsyCHIC), université de Lille, 59000 Lille, France.
REVIEW RETURNED	01-Mar-2018

GENERAL COMMENTS	Thank you for giving me the opportunity to read and comment this manuscript. The authors are to be commended on such an important and well-conducted piece of work. I look forward to seeing this article in press in BMJ Open. The work reported here has clinical relevance and has potential important implications for health policy in prisons. The treatment pathway proposed for insomnia management in prison is very interesting. The issue of the feasibility of setting up such a protocol in prison arises but is clearly discussed by the authors in the article. The Delphi method used in this paper appears particularly suited to the targeted problem. Indeed, it allows a multidisciplinary approach adapted to take into account all aspects of a complex environment such as prison. I have only a few minor comments and suggestions that could be addressed before publication. Abstract A brief description of what a Delphi technique could be given in the design section. Introduction The context of this study is clearly described. Methods The authors should explain why the Delphi technique is “modified” here. They should cite other studies in which a Delphi technique with 3 rounds was used too. It could be particularly interesting for the reader, to give the main characteristics of the stakeholders (academic sleep researchers, prison staff and prisoners). Particularly, in which type of prison the
--

	staff and prisoners were recruited? Results It should be explained why only 13 stakeholders joined round 2 and 3, instead of 15 in round 1? Discussion It is hard to understand why the Sleep Condition Indicator appears in the Discussion section whereas it is not mentioned in the results neither in the Figure 2. Please clarify. Figure 2 The color code is hard to read in this figure and the meaning of the different shapes used (circle, diamond, etc.) is not explained.
--	---

REVIEWER	Ralitsa Akins US, WSU, ESFCOM
REVIEW RETURNED	28-Mar-2018

GENERAL COMMENTS	Thank you for this opportunity to review your work. I was very enthusiastic about your topic and eager to read about your approach. There are a number of considerations, and the main one for me was the stated study aim, "to design an evidence-based treatment pathway for insomnia, acceptable to staff and prisoners" and its dissonance with the described approach and results, that did not support exactly this aim. The study supported the discovery of whether a suggested pathway was acceptable to panelists, including researchers, prison administrators and prisoners. The study did not support the evidence-based way of developing said pathway, which was, at best, insufficiently mentioned and apparently a result of previous studies. Hence, the stated study aim was over-reaching and need be reviewed. In Delphi studies, the collective expertise of the participants will resonate on the soundness of the consensus reached. Your participants did have an expertise whether a certain pathway for addressing insomnia in the reality of imprisonment would be acceptable, yet collectively the panelists did not have expertise on the evidence-based nature of the developed pathway. Therefore, your study was about how such treatment may be felt about/viewed and/or adhered to by prisoners, and about its feasibility in said environment, rather than an evidence-based treatment of insomnia. Panelists, unless trained in development and application of insomnia therapy, cannot be assumed to be holders of evidence-based approaches of effectiveness of treatment for insomnia, and your panel was widely diverse in such expertise, or lack there of. In light of the aforementioned, I would strongly suggest, that you revise and clarify the aim and outcomes of your study, to reflect the expertise of the panel. Several related matters involve:  - In the text you talk about "2 researchers" and in the table there are "3" mentioned. Please, clarify and align. - While important study limitations are acknowledged, you claim to have developed (p.10, lines 46-47) a "comprehensive treatment pathway" - please clarify and align this overconfident statement. - On p.10, paragraph 1, lines 3-9, you "hypothesise the pathway would positively impact upon healthcare utilisation rates and help reduce sedative-related prescriptions." Such a statement seems over-reaching and not supported by your study outcomes. The base for such hypothesis is not established, and it would be prudent to
---

	avoid unsupported statements.  - It is not clear whether the limited number of prisoners-participants have been treated for insomnia in prison, and where they draw their opinions on. Would be good to clarify in detail how recruitment of vulnerable populations to this study was completed in light of available qualified participants, and the possible bias in limited sample selection. - Would be important to describe/state what were the qualifying expert populations for this study, how the recruitment was done for each sub-population, and the percentage of individuals who agreed to participate on the panel(s) out of the available qualifying populations. - Without clear criteria for sample selection, the results of the study would carry a possibly important bias and adversely affect any consensus reached. - I was concerned about not obtaining an IRB/ethical clearance when using vulnerable populations in research, despite the presumed "service" nature of the project. It was not clear if prisoners-participants were hoping for some unwritten benefits by participating in this study, or gaining a more favorable status by prison administration. While I understand the designation of this study as "service", my personal recommendation would be to always seek ethics review rather than skip it. While interesting as a concept, your study would need better alignment between expertise of panelists and drawn conclusions. I am looking forward to reviewing an updated and aligned version.
--	--

VERSION 1 – AUTHOR RESPONSE

Reviewer 1

Abstract

A brief description of what a Delphi technique could be given in the design section.

The following has been added to the design section of the abstract: "A modified Delphi technique, adapted to the stakeholder (either receiving controlled feedback online or face-to-face on a series of statements), was used over three rounds to gain consensus on a final treatment pathway design".

Methods

The authors should explain why the Delphi technique is "modified" here.

*We have now made it clear how the Delphi technique was modified on page 5. A sentence has been changed (in bold) to reflect this in the methods section: "Traditionally in a Delphi study experts do not know each other, do not work or live together and usually respond to consensus rounds via email or other electronic means. This was not possible for prisoners **therefore the Delphi technique was modified; due** to a lack of access to electronic communication in prison and, to fit within the resources of the study, the lead author [LD] met with a group of women prisoners together in prison over a series of three face-to-face meetings."*

| They should cite other studies in which a Delphi technique with 3 rounds was used too.

The following has been added to the first paragraph in the method section on page 5: "There is no mandated number of rounds required to complete a Delphi consultation; the process is repeated until

consensus is achieved. However, having at least two rounds of feedback is commonplace, with some studies using three [19-21].

It could be particularly interesting for the reader, to give the main characteristics of the stakeholders (academic sleep researchers, prison staff and prisoners). Particularly, in which type of prison the staff and prisoners were recruited?

The following has now been added to the first paragraph of the results section on page 6: "Two academic sleep researchers, three prison healthcare staff (two from a male prison holding unconvicted men and those serving shorter sentences and one male prison for these part way through longer sentences) and six prisoners (from a female prison) reviewed the initial pathway".

Results

It should be explained why only 13 stakeholders joined round 2 and 3, instead of 15 in round 1?

The majority from round 1 responded in round 2 (n=13; 87%); one prisoner did not continue to round 2 and 3 because they no longer wanted to be involved, and the GP did not continue because of unprecedented work commitments.

Discussion

It is hard to understand why the Sleep Condition Indicator appears in the Discussion section whereas it is not mentioned in the results neither in the Figure 2. Please clarify.

The first paragraph of the results section has been revised to include the following: "Two academic sleep researchers, three prison staff and six prisoners reviewed the initial pathway. It structured to have a short screen for insomnia, using the Sleep Condition Indicator (SCI) [27], initial assessment (potential causes of insomnia, sleep care plan etc.), and short-term (medication) and long-term treatment (individual components of CBTi) options."

Figure 2

The color code is hard to read in this figure and the meaning of the different shapes used (circle, diamond, etc.) is not explained.

The colour code on Figure 2 has been revised. It now includes explanation of the different shapes.

Reviewer 2

There are a number of considerations, and the main one for me was the stated study aim, "to design an evidence-based treatment pathway for insomnia, acceptable to staff and prisoners" and its dissonance with the described approach and results, that did not support exactly this aim. The study supported the discovery of whether a suggested pathway was acceptable to panelists, including researchers, prison administrators and prisoners. The study did not support the evidence-based way of developing said pathway, which was, at best, insufficiently mentioned and apparently a result of previous studies. Hence, the stated study aim was over-reaching and need be reviewed.

Thank you for your useful comment. We have revised the aim accordingly. The objective within the abstract now reads: "The study aim was to design a treatment pathway for insomnia in prisons informed by stakeholders with professional or lived experience of insomnia and prison-based interventions". The aim referenced in the introduction is now reads: "This study employed a modified Delphi study to design a treatment pathway for insomnia in prisons informed by stakeholders with professional or lived experience of insomnia and prison-based interventions".

In Delphi studies, the collective expertise of the participants will resonate on the soundness of the consensus reached. Your participants did have an expertise whether a certain pathway for addressing

insomnia in the reality of imprisonment would be acceptable, yet collectively the panelists did not have expertise on the evidence-based nature of the developed pathway. Therefore, your study was about how such treatment may be felt about/viewed and/or adhered to by prisoners, and about its feasibility in said environment, rather than an evidence-based treatment of insomnia. Panelists, unless trained in development and application of insomnia therapy, cannot be assumed to be holders of evidence-based approaches of effectiveness of treatment for insomnia, and your panel was widely diverse in such expertise, or lack thereof.

Again, thank you for your observation. The pathway itself was, as much as possible, based on NICE treatment standards for insomnia and evidence of the effectiveness of treatments. This includes but not limited to, the inclusion of cognitive behavioural therapy for insomnia (CBTi), the gold standard treatment for chronic, long-term insomnia, and hypnotic medication for short-term insomnia; these treatment options were verified by the academic sleep stakeholders, with training in development and application of insomnia treatments. However, we acknowledge that the majority of the stakeholders did not have this expertise. Nevertheless, we, as a research team, have expertise in insomnia in prisons both by knowledge of our initial findings and expertise by experience in working within the forensic mental health sector and service provision.

That being said, we have removed “evidence-based” from the aim and abstract.

In light of the aforementioned, I would strongly suggest, that you revise and clarify the aim and outcomes of your study, to reflect the expertise of the panel.

We have revised the aim and outcomes as per the above answer.

Several related matters involve:

- In the text you talk about "2 researchers" and in the table there are "3" mentioned. Please, clarify and align.

Thanks for your comment. In the manuscript, it currently reads; “Two academic sleep researchers, two healthcare managers and one GP and three prisoners from the initial group continued into the Delphi rounds and were joined by additional relevant stakeholders”. This means the two academic sleep researchers, from the initial group, continued into the Delphi round and an additional academic sleep researcher joined them at this stage. An additional column has been added to Table 1 to make this clearer.

Table 1: Retention of stakeholders by meeting and Delphi round

Stakeholder	Initial group meeting	Delphi		
		Round 1	Round 2	Round 3
Academic sleep researcher	2	3	3	3
Prisoner	6	3	2	2
GP	1	2	1	1
Healthcare manager	2	2	2	2
Primary care manager	–	1	1	1
Psychologist	–	1	1	1
IAPT staff member	–	1	1	1
Mental health lead	–	1	1	1

Prison governor	–	1	1	1
Total	11	15	13	13

- While important study limitations are acknowledged, you claim to have developed (p.10, lines 46-47) a "comprehensive treatment pathway" - please clarify and align this overconfident statement.

The authors identify the pathway as comprehensive because all aspects of possible treatment regimens were included or considered for inclusion including but not limited to self-help, peer support, medication and CBTi. The existing sentence clarifies this: "We have produced a comprehensive treatment pathway for insomnia with an emphasis on self-management, peer support and psychological approaches, further modified by consensus across sleep academics, healthcare professional and prisoner stakeholders." It was also described as comprehensive by stakeholders on page 6 of the results section.

- On p.10, paragraph 1, lines 3-9, you "hypothesise the pathway would positively impact upon healthcare utilisation rates and help reduce sedative-related prescriptions." Such a statement seems over-reaching and not supported by your study outcomes. The base for such hypothesis is not established, and it would be prudent to avoid unsupported statements.

Thank you for your comment. When the treatment pathway is piloted in a prison, insomnia issues and the number of prescriptions will be an outcome of interest for the prison healthcare department. As such, we feel this statement should remain as a likely outcome in the next study.

The following has been added to the unanswered questions and future research section of the discussion on page 11: "In addition, the number of consultations for sleep problems and sedative-related prescriptions would be outcomes of interest".

- It is not clear whether the limited number of prisoners-participants have been treated for insomnia in prison, and where they draw their opinions on. Would be good to clarify in detail how recruitment of vulnerable populations to this study was completed in light of available qualified participants, and the possible bias in limited sample selection.

Thank you for your useful comment. Firstly, prisoners, were identified as service users and not participants. They all had lived experience of insomnia and had had previously asked for help for sleep problems in the prison setting; indeed, they were aware of how it was currently managed and could inform on potential realistic changes. The Deputy Governor of a women's prison was approached to see if we could attend one of the existing monthly prisoner health representatives' consultation meetings. The lead author [LD] attended a meeting, presented information about the service improvement project and asked if anyone wanted to be involved in reviewing and commenting on a treatment pathway diagram of insomnia in prison. They had not been a participant in one of the other studies.

The following has now been added to the method section on page 5: "A monthly prisoner health representatives' consultation meeting was identified as a way to purposively identify potential stakeholders. This group had lived experience of insomnia in prison, had previously asked for help for sleep problems in prison and were aware of the range of treatments currently available in prison. Subsequently, the lead author felt, overall, this collective expertise was adequate to proceed with the initial review of the pathway."

- Would be important to describe/state what were the qualifying expert populations for this study, how the recruitment was done for each sub-population, and the percentage of individuals who agreed to participate on the panel(s) out of the available qualifying populations. - Without clear criteria for sample selection, the results of the study would carry a possibly important bias and adversely affect any consensus reached.

Thanks for your comment. We used a convenience sample of stakeholders as mentioned in the manuscript. Academic sleep researchers were approached because the lead author [LD] was aware of their expertise in both managing insomnia as a healthcare professional and their research that examined insomnia treatment. Specific prison staff were approached because they had experience in delivering, implementing and evaluating prisoner health issues, including sleep problems. Qualifying expertise for service users is discussed above. Despite acknowledging expertise in the manuscript, the level of expertise was not robustly determined before stakeholders were approached. We also state this as a limitation in the strengths and weaknesses of the study section of the discussion.

The following has been added to the method section on page 5: "Stakeholders were invited to contribute based on their professional or lived experience of insomnia and prison-based interventions, and availability: academic sleep researchers, because the lead author [LD] was aware of their expertise in both managing insomnia as a healthcare professional and their research that examined insomnia treatment, and specific prison staff because they were had experience in delivering, implementing treatments and evaluating treatments for prisoner health issues, including sleep problems.

All academic sleep researchers (n=2) and prison staff (n=3) who were approached agreed to participate on each panel. Just under two-thirds (60%) of the representatives at existing health consultation meetings agreed to consult on the project. However, it is possible the other representatives did not have experience of having a sleep problem in prison.

- I was concerned about not obtaining an IRB/ethical clearance when using vulnerable populations in research, despite the presumed "service" nature of the project. It was not clear if prisoners-participants were hoping for some unwritten benefits by participating in this study, or gaining a more favorable status by prison administration. While I understand the designation of this study as "service", my personal recommendation would be to always seek ethics review rather than skip it. While interesting as a concept, your study would need better alignment between expertise of panelists and drawn conclusions. I am looking forward to reviewing an updated and aligned version.

Thank you for your comment and advice. It was clearly explained to the service users in the three consultations that there were no unwritten benefits by commenting and consulting on the pathway design document. They were specifically told that they were advising on service planning for insomnia treatment in prison and were not participants in a research study. As such ethical approval for the specific study was not necessary.

The design of the treatment pathway was stage 5 of a larger doctoral grant that was included in the IRAS ethics application; ethical approval was gained for the overall study (references 3, 5, 14, an unpublished qualitative study and the design of the pathway). However, the specific design of the pathway stage was specifically service planning/evaluation. We also used the tool: <http://www.hra-decisiontools.org.uk/research/> and it confirmed it was not research. We followed the same rules as anyone would when involving patients in service planning and evaluation in non-prison establishments and community where you would never need ethics; process equivalence adhered to community practice. Furthermore, including service users as advisors is a common model in prison and the subject matter did not bring with it any specific considerations of additional vulnerability.

The following has been added under the "Ethical approval" section: "The design of the treatment pathway was the last stage of a five-study large doctoral grant that was included in the IRAS ethics application; approvals from NHS (REC for Wales; reference: 13/WA/0249) and HM Prison Service ethics committees (NOMS; reference: 2013-208) were obtained. However, the design of the pathway stage was specifically identified as a service improvement project and thus did not require individual ethical approval."